# Engaging in and Sustaining Physical Activity and Exercise: A Descriptive Qualitative Study of Adults 65 Years and Older Using the Self-Determination Theory

Anittha Mappanasingam [1], Katelyn Madigan [1], Michael E. Kalu [2], Melody Maximos [3] and Vanina Dal Bello-Haas [1,*]

1 School of Rehabilitation Science, McMaster University, 1400 Main Street West, Hamilton, ON L8S 1C7, Canada; mappanaa@mcmaster.ca (A.M.); katelyn92011@hotmail.ca (K.M.)
2 School of Kinesiology and Health Science, Faculty of Health, York University, 328 Stong College, 165 Campus Walk, 47000 Keele Street, Toronto, ON M3J 1P3, Canada; mkalu@yorku.ca
3 CBI Health, 110-116 Greenbank Road, Nepean, ON K2H 5V6, Canada; maximosmelody@gmail.com
* Correspondence: vdalbel@mcmaster.ca or vaninadbh@gmail.com; Tel.: +1-289-775-2114

**Abstract:** Introduction: Physical activity (PA) and exercise (EX) participation rates have not been increasing among older adults, with many not meeting recommended guidelines. This qualitative descriptive study examined factors influencing engagement in PA within and outside an older adult fitness club context, using self-determination theory (SDT). Methods: Thirty-seven community-dwelling adults 65 years and older participated in focus groups or telephone interviews. Two researchers independently coded and analyzed transcript data inductively and deductively using SDT. Results: Two broad themes, *The Spectrum of Motivating Factors* and *Facilitators and Barriers*, and nine sub-themes, *Physical Activity and Exercise Brings Me Joy; Meaningful Personal Impetuses; I Get Active with a Little Help from my Spouse and Others; I See Changes and Improvements* (Theme 1); *I Can Do This; Connections and Sense of Belonging; I Cannot Do This; Setting, Environment, and People Supports;* and *Pragmatics* (Theme 2), emerged from the data. All participants discussed several motivating factors: enjoyment, managing health conditions, being held 'to account' by others, opportunities for socialization, and seeing improvements in health and well-being. A lack of supportive environments, knowledgeable staff and suitable settings and programs were cited as barriers by participants who were not older adult fitness club members. Discussion: Factors along the extrinsic to intrinsic regulation continuum facilitated or hindered community-dwelling older adults to engage in and sustain PA within and outside an older adult fitness club context. The findings underscore the need for programs, settings, environments, and related components to be expressly older-adult-tailored to enhance motivation through competence, autonomy, and relatedness support for maximal engagement and participation in PA or EX.

**Keywords:** older adults; physical activity; exercise; motivation; self-determination theory; qualitative descriptive study; hybrid inductive–deductive thematic analysis

## 1. Introduction

With a rapidly growing older adult population, which is expected to double from one billion in 2020 to 2.1 billion in 2050 [1], concerns exist about health care systems being adequately prepared with sufficient knowledge and resources to support adults across their life course in the development and maintenance of functional ability that enables well-being [2,3]. The World Health Organization's Guidelines on Integrated Care for Older People (ICOPE) [4] has proposed evidence-based recommendations to prevent, slow, or reverse declines in the physical capacities of older people, including guidance on physical activity (PA)—any bodily movement produced by skeletal muscles that results in energy expenditure [5] and multimodal exercise (EX)—a subset of PA that is planned, structured,

and repetitive with the objective to improve or maintain one or more components of physical fitness [5]. Recommendations for individuals 65 years of age and older include: 150 to 300 min of moderate-intensity aerobic PA or 75 to 150 min of vigorous-intensity aerobic PA throughout the week; muscle-strengthening activities at a moderate or greater intensity involving all major muscle groups two or more days per week; and varied multicomponent PA or EX that emphasizes functional balance and strength training at a moderate or greater intensity three or more days per week [6].

A lack of PA is the fourth leading risk factor for global mortality causing approximately three million deaths per year [7,8]. Conversely numerous benefits of PA or EX have been well-established over the years: (1) physical health benefits, e.g., weight loss and maintenance, chronic disease management [9]; (2) mental well-being benefits, e.g., stress relief, decreased depression symptoms [9]; (3) improved cognitive function, e.g., improved memory and spatial learning [10–13]; (4) chronic disease risk reduction, e.g., cardiovascular disease and hypertension [14,15]; (5) decreased risk of falls [16–18]; and (6) improved physical function, physical independence, and mobility [16,18–20].

Despite reported risks of inactivity and the numerous benefits of PA, participation rates have not been increasing in the older adult population. Many older individuals, including those with chronic conditions, are not meeting recommended guidelines [7,21,22], and some older adults, despite enrolling in a program, struggle to maintain PA levels or meet the minimum recommendations [23]. For health care and other exercise professionals to best promote PA practices, guide and support sustained engagement, and assist community-dwelling older adults to meet the PA frequency, time, type, and intensity recommendations, it is important to understand the motivational factors that may facilitate or hinder regular and sustained engagement.

The literature regarding factors influencing participation in PA in the older adult population is largely quantitative in nature, while studies examining older individuals' motivations using qualitative methodology that also incorporates theory are less common. Few studies have included "older" adults participants [24]—this is especially important as motivational factors may vary with age [24] and the number of people 80 years and older is expected to triple to 426 million by the year 2050 [1]. Similarly, studies exploring the potential influences of belonging to a fitness club specifically geared towards older adults are limited. A 2018 study found older adults randomized to 65+ years group-based EX programs had greater adherence rates compared to older adults randomized to a standard group-based EX program that included older and younger individuals, suggesting age-targeting programs may be beneficial [25]. An older adult fitness club context may provide distinct advantages important for supporting and enhancing older adult engagement in PA.

Over the past two decades, the value and relevance of self-determination theory (SDT, Figure 1) in understanding PA behavior [26] has been underscored, with increasing applications to the health domain, health care, and health behavior intervention research [27–29]. SDT, a broad theory of human behavior, conceptualizes how behavior is influenced by personal and contextual motivational factors [30,31]. Motivation is considered a multi-faceted construct comprised of regulatory styles along a continuum of relative autonomy or self-determination. SDT describes the way intrinsic motivation, e.g., internally regulated and based on the self (desire, growth), and extrinsic (externally regulated, varying degrees of autonomy) motivation, e.g., reward, punishment, and guilt, can help individuals engage in behaviors and activities [30,31].

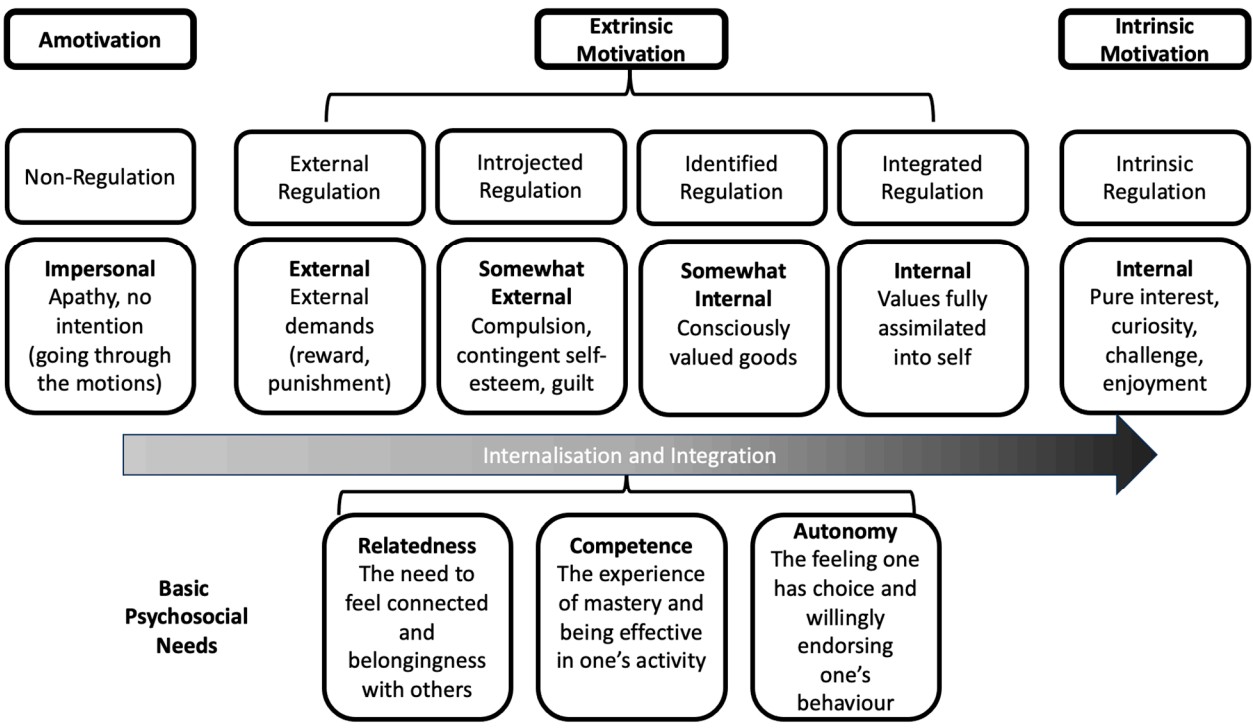

**Figure 1.** Self-determination Theory [30–32].

The fulfillment of three psychological needs relates to and is critical to motivation: (1) competence—the ability to gain new skills and control what we do, and makes us feel good about the tasks we have completed; (2) relatedness—the need to connect and interact with others, establish meaningful relationships, and feel a sense of belonging; and (3) autonomy—the ability to make our own choices, related to the need to feel our behavior is voluntary and self-endorsed [30–32]. These three needs being met leads to internalization, engagement in the behavior becoming autonomously regulated and valued over time, and persistence with the behavior [30,31]. Ongoing supports for the three psychological needs facilitate an individual's development and engagement in and integration of a behavior [31,32]. Environments that support autonomy (an autonomy-supportive environment) in SDT theory can impact motivation, wellness, and performance. Similar supports (supportive environments) for relatedness and competence can interact with the decision making related to motivation and engagement. The interaction between the individual and social context that provides support (or not) can influence and impact the three psychological needs, and in turn motivation [32].

The overall aim of this qualitative study was to explore and describe the factors that influence older adults to engage in and sustain PA, within and outside an older adult fitness club context, using the SDT [30,31] to frame the research. The research questions addressed were: (1) what are the perspectives and experiences of adults aged 65 and older regarding PA within and outside the context of an older adult fitness club? And (2) what factors influence adults ages 65 years and older to engage in and sustain PA, within and outside the context a fitness club for older adults?

## 2. Materials and Methods

This was a qualitative descriptive [33,34] study, with a constructivism [35] lens, of community-dwelling adults 65 years and older living in a metropolitan city in Ontario, Canada. Qualitative description research tradition is often used to explore, describe, and develop deeper understanding of health and health-care-related phenomena and participant views and experiences to inform practice [36,37]. A constructivism lens acknowledges that individuals develop subjective meanings of their experiences through seeking to un-

derstand the world in which they live [35]. We followed the Standards for Reporting Qualitative Research (SRQR) [38] to guide the reporting of this study.

## 2.1. Study and Sampling Context

This study was conducted in a city in Canada (name withheld for anonymity purposes). Census data available at the time of the study classified the city as a large urban population center, with adults 65 years and older comprising 17.3% of the population (5.5%—65 to 69 years; 6.9%—70 to 79 years; 3.9—80 to 89 years; 1.0% 90 years and older); 44.2% of the older adult population were males and 55.8% were females; and, 11.6% had a low income (Low Income Measure) [39]. *Ethnic origin (*Canadian Census wording/categories) composition was: 2.71% First Nations; 0.04% Inuit; 0.7% Métis; 5.8% South and Southeast Asian, 4.4% South Asian, 4.0% West Central Asian and Middle East, 2.7% African, 2.4% Caribbean, 2.2% Latin, Central, and South American, 0.2% Oceania, 25.5% other North American, e.g., Canadian or Acadian; and 72.5% European [39]. Education and living situation were not available solely for individuals over 65 years.

The city offered a variety of community-based "Older Adult 55+" PA, i.e., dance, and EX, i.e., strengthening programs. Program length varied and fees and physician clearance requirements were dependent on program type. The older adult fitness club, from which a portion of the purposeful sample was drawn, is housed at a non-profit, multi-level, and multi-service residential facility (e.g., apartments, assisted living, long-term care, convalescent care, day programs). The fitness club is open to facility residents, with membership being automatic. Community-dwelling individuals over the age of 65 years residing in the city and surrounding area may also become members. Club members require physician clearance to join, pay CAD 25 for an initial assessment conducted by staff, and the membership fee is a renewable four-month membership (CAD 40/month). An individualized program was developed based on the initial assessment and member goals, and one-on-one support was provided as needed.

## 2.2. Sample Participants and Recruitment

Criterion sampling, a purposeful sampling strategy used to identify and select information-rich cases meeting predetermined criteria deemed important to address the study aim and research questions [40], was used. A purposeful sample of three participant groups were recruited: current members of the older adult fitness club, individuals engaged in community-based PA, and individuals interested in engaging in PA, but not doing so at that time. Recruitment methods were varied to target the three groups: posting and distribution of ethics-approved advertisements to a wide number of organizations and centers that offered community-based health, recreation, social, and other services for older adults; posters at the older adult fitness club; newspaper advertising; and social media.

Participants were eligible to participate if they were at least 65 years of age, independently living in the community, and able to communicate in English. Determining an adequate sample size for qualitative research studies is dependent on many factors. As specific parameters are lacking, the most common guiding principle for sample size is saturation [41,42], with the most frequently reported sample sizes in qualitative descriptive studies using interviews or focus groups ranging from 8 to 30 participants [36]. In keeping with qualitative research methodology guidelines, no a priori sample size was determined, and we aimed to initially enroll 10 participants in each participant group type. We were prepared to continue enrollment beyond this number until code saturation (no new or additional codes were identified in the data) and meaning saturation (no new insights or understandings were identified in the data) were reached [41,42].

## 2.3. Data Collection

Participants were offered the choice of an individual telephone interview (30 to 45 min in length) or a focus group (45 to 60 min in length). Focus groups comprised of older adult fitness club members (Focus Group 1, *n* = 9; Focus Group 2, *n* = 6) or older adults

participating or interested in participating in PA or EX in the community (Focus Group 3, *n* = 5; Focus Group 4, *n* = 9; Focus Group 5, *n* = 6). All but two (both older adult fitness club members) individuals participated in one of the focus groups.

A research team member initially provided participants with an overview of the study and what was involved. After answering questions about the research and interview or focus group processes and prior to completing the interview or focus group, participants completed an informed consent form and a questionnaire that collected demographic and other participant data, such as number of chronic conditions, use of assistive devices, and physical functional status. The consent form and questionnaire were sent to interview participants in advance and returned prior to the phone call interview. Focus group participants completed consent forms and questionnaires in person, on-site. To elucidate physical functional status, participants completed the Composite Physical Function Scale (CPFS), a 12-item scale with evidence of validity and reliability in older adults [43,44]. CPFS items include: basic activities of daily living, e.g., personal care, bathing; walking activities, e.g., one-to-two blocks, ½ mile, one mile; instrumental activities of daily living, e.g., shopping for groceries or clothes; heavy household activities, e.g., scrubbing floors, vacuuming, raking leaves; and strenuous activities, e.g., hiking, moving heavy objects. Items were scored as 0 (cannot do), 1 (can do with difficulty or help), and 2 (can do). Scores range from 0 to 24, with high functioning considered a score of 24; moderate functioning is considered a score of 14 (ages 90 and above), 16 (ages 80–89), 18 (ages 70–79), or 20 (ages 60 to 69). We also asked participants the amount they would spend for PA and EX on a per month basis, e.g., monthly membership/program fee and individual class/session basis.

All interviews and focus groups were conducted by at least two members of the research team. The interviews were conducted via speaker phone and one team member served as facilitator and another took field notes. For each focus group there was one facilitator and two observers who assisted the participants with reviewing and completing the consent form and questionnaire as needed and who took notes. The research team member who conducted the interview or served as focus group facilitator was experienced in these methods—they had related formal education and informal training and had served in these roles for several research projects. All research team members involved underwent training and made reflexive notes during and after the interview or focus group sessions. Focus groups and interviews were digitally recorded, and recordings and field and reflexive notes were transcribed verbatim and de-identified.

A semi-structured interview guide was used, comprised of the same introductory information and similar (e.g., wording slightly different for older adult fitness club members' focus group) open-ended and probing questions related to perspectives and experiences with PA or EX in general; perspectives and experiences with the fitness club or PA or EX programs; reasons for (or not) engaging in or sustaining PA or EX or a program in the community or via the fitness club; and perceived strengths and areas for improvement for the fitness club or community-based PA or EX programs and activities. The interview guide was developed iteratively from a review of the published literature and similar research, and through discussions and revisions by the research team members. The two interviews were conducted first, and the final question asked the participants if they felt there were any questions that should have been asked or if there was anything else they would like to discuss that was not asked. Team members debriefed after each interview and focus group to determine if any changes were needed.

*Reflexivity Statement.* The broader research team included researchers, research staff, and trainees in health professional or health-related programs. Team members involved in data collection and analysis were mostly females, ranging in age from mid-twenties to early sixties, had a variety of ethnic and cultural backgrounds, and had personal, professional, and research experience with older adults, people living with chronic health conditions and disabilities, and PA. The research team had been collaborating with administrators and staff at the facility housing the older adult fitness club on research and program evaluation initiatives for several years, but did not have any relationship with the participants. As

part of the introduction to the research, processes, and procedures, team members briefly introduced themselves and their roles, e.g., researcher, research coordinator, student (graduate or undergraduate), research assistant, or health professional, as applicable, but did not provide specific reflexivity statements.

*Ethical Considerations.* Ethics approval was received from the required Institutional Research Ethics Board (HiREB #5026). Participants were provided with information about the study and signed a consent form prior to participation. Demographic forms included a participant identification number and had no identifying details. Any specifics within the transcripts were removed or de-identified for anonymity, e.g., participant names were removed and identification numbers were assigned to transcripts; potentially identifying information within the recordings were removed or replaced with pseudonyms or XXX.

*2.4. Analysis*

Demographic and other participant variables were analyzed using descriptive statistics [Statistical Packaging for Social Sciences, SPSS 28.0]. Transcript data were analyzed using a hybrid inductive–deductive [45,46] thematic analysis approach to identify, examine, organize, describe, and report patterns (themes) found within the data that emerged as being important to the description of the phenomenon. Inductive coding began with observations of the data and sought to find a pattern within the data to generate themes—codes and categories were not predetermined and were identified through reading the data [45,47,48]. Deductive coding involves applying pre-determined categories and codes based on the research purpose or questions or as generated from the literature or from theory to the data [45,46]. SDT [30,31] and related constructs and concepts were used as categories exploring if and how these categories fit with and explained the data. Combining inductive and deductive analysis harnesses the value and advantages of each [49].

Two researchers independently conducted an iterative inductive and deductive cycle analysis process, as described by Bingham and Witkowsky [50], keeping reflexive notes and memos, e.g., reasons for codes, connections between codes, and arising subthemes: (1) Initial impressions and familiarization with the data by reading and re-reading the transcripts. (2) Generation of initial codes that are interesting and meaningful via line-by-line reading and coding of the data. Researchers referred to observations, interviews, and focus group notes to further contextualize the codes as needed. (3) Searching for themes and relationships—codes were sorted and classified into themes and sub-themes. (4) A deeper review of themes and relationships—themes were modified as needed and a preliminary set of themes and sub-themes were generated. Researchers then met to discuss their coding and interpretations of meaning. (5) Final analysis—further refinement and data triangulation to define (name) the themes and sub-themes. Meetings with the broader research team not involved in the coding process were held throughout the data collection and preliminary data analysis steps, and final analysis and findings were presented for review and feedback. Best practice for rigor, trustworthiness, and quality analysis were implemented, e.g., two researchers conducting analyses independently, audit trail, data collection triangulation, researcher triangulation, and peer debriefing [36,45,48].

## 3. Results

The 37 participants, with ages ranging from 65 to 92 years, included: 17 older adult fitness club members, 8 participants who were engaged in community-based PA or EX, and 12 who were interested in engaging in PA or EX. The CPFS scores ranged from 0 to 24, mean = 19.36 (SD = 6.03). The mean CPFS score by age group was indicative of high functioning, except for participants 60 and 69 years of age, with a mean score just below the cut-off (19.4, SD = 4.16). A majority (82.4%) of the older adult fitness club members were living in the community and not in one of the facility's apartments, and the club membership length ranged from 3 months to 12 years (mean = 5.23 years, SD = 4.83). (See Table 1 for additional participant characteristics and information).

**Table 1.** Participant Characteristics (*n* = 37).

| Variable | *n* (%) OR Mean (SD), Range OR Mode, Range |
|---|---|
| Sex, female | 23 (62.2%) |
| Age, years | |
| Older adult fitness club member participants | 80.9 (7.4) |
| Participants engaged in community-based PA | 77.7 (8.0) |
| Participants interested, but not engaged in community-based PA | 76 (SD 6.3) |
| Ethnic origin | |
| Caribbean | 3 (8.1%) |
| North American | 18 (48.6%) |
| European | 16 (43.2%) |
| Education Level | |
| Did not complete high school | 7 (16.2%) |
| Completed high school degree | 6 (18.9%) |
| Completed a college or university degree | 20 (54.1%) |
| Other training | 3 (8.1%) |
| Living Status | |
| Lives alone | 19 (51.4%), |
| Lives with spouse or partner | 14 (37.8%) |
| Lives with family members or friends | 4 (10.8%) |
| Employment status | |
| Retired, not working | 35 (94.6%) |
| Working part-time | 1 (2.7%, participant aged 73 years) |
| Working full-time | 1 (2.7%, participant aged 87 years) |
| Number of health conditions * | 2, 0 to 8 |
| Assistive device use | |
| Walker | 3 (8.1%) |
| Cane | 4 (10.8%) |
| Scooter | 1 (2.7%) |
| Wheelchair | 2 (5.4%) |
| Amount participant willing to pay for PA or EX | |
| An individual PA or EX session | |
| CAD 3 | 32 (86.5%) |
| CAD 5 | 5 (13.5%) |
| Membership or program fee | |
| CAD 25 to CAD 50/month | 32 (86.5%) |
| CAD 50 to CAD 75/month | 3 (8.1%) |
| As low as possible | 2 (5.4%) |

PA = physical activity; SD = standard deviation; * Reported health conditions were varied and included: cancer, multiple sclerosis, asthma, peripheral vascular disease, Parkinson's disease, heart disease, lung disease, diabetes, neuropathy, hypertension, stroke, osteoporosis, rheumatoid arthritis, osteoarthritis, and epilepsy.

### 3.1. Themes and Sub-Themes

The two main themes, *The Spectrum of Motivating Factors* and *Facilitators and Barriers*, their related subthemes, and linkages to SDT [30,31] concepts are described below, with representative quotes.

Theme 1: *The Spectrum of Motivating Factors*

A variety of motivating internal and external stimuli along the regulation continuum were described as important reasons and prompted the older adults to initially engage in PA or EX, make decisions about participation, work towards achieving their goals, and sustain their engagement. No participants referred to situations or described themselves in

such a way that would signify motivation. There were four sub-themes within this theme: *Physical Activity and Exercise Brings Me Joy; Meaningful Personal Impetuses; I Get Active with a Little Help from my Spouse and Others;* and *I See Changes and Improvements*.

### 3.2. Physical Activity and Exercise Bring Me Joy

Many participants described being intrinsically regulated through their expressed feelings of enjoyment, excitement, and personal accomplishments. They were genuinely interested in, gained satisfaction from, and enjoyed and found pleasure in engaging in PA.

They either had always derived inherent gratification and satisfaction from engaging in PA or EX or this developed over time . . .

> "I do [enjoy participating in exercise]. I like my swimming." . . . "You know just getting there, once you get there you feel I did it, I feel good now... You know...it's a good feeling when you've worked out hard."
>
> (C-FG3-P3)

> "When I was younger I was a fairly good athlete and I was very competitive. That never quite goes away. So now I compete with myself." . . .it's a good feeling when you've worked out hard."
>
> (C-FG3-P2)

> "It feels good after you do-I do it every morning."
>
> (FC-FG2-P4)

### 3.3. Meaningful Personal Impetuses

Participants described several personal value- and utility-related factors, indicative of more autonomous regulation. The importance of maintaining and improving overall fitness, physical and mental health and well-being, mobility, and function through PA, and wanting to achieve specific outcomes such as maintaining or losing weight or improving balance, strength, mobility, and flexibility were cited as reasons for participating in PA:

> "lose weight...I'd like to build up more muscular strength...flexibility is critical as we get older."
>
> (C-FG3-P1)

> "And just to minimize the risks. You know and...like because I'm not flexible and not doing anything so I don't...so you know bending down is much harder now with my knees. So I try to do little things like look at my balance when I'm...you know standing on one leg."
>
> (C-FG3-P4)

> "My goal was to walk without a friend here (referring to their walker)."
>
> (FC-FG1-P6)

> "I already had a problem with balance and walking so for me it was just an automatic thing to try and get help to try and get back on my feet . . . I wanted to get rid of my walker..."
>
> (FC-FG1-P4)

Participants also discussed health condition management as an important motivator. Although discussed by all participant groups, this was particularly evident among the older adult fitness club members. Being diagnosed with a health condition, such as diabetes and arthritis, and wanting to improve and maintain their current level of function and abilities, whether these were affected by a particular health condition, were described as essential reasons for undertaking PA:

> "....arthritis, try to keep it at bay. And avoiding surgery."
>
> (FC-FG1-P5)

"I had a lot of pain and I couldn't sleep."

(FC-IN2-P1)

"I'm Diabetic. And getting off my duff is kind of important and it's hard to do."

(C-FG3-P1)

Taking control of their health and well-being, and addressing changes associated with aging or their specific health conditions were their choices and volitional decisions. These participants were originators of their actions, demonstrating autonomy and autonomous decision making. Conversely, situations and factors more indicative of controlled regulations were also described as providing a meaningful impetus to engage in PA. For some participants, it was a physician or other health care professional who initially recommended they engage in PA to address a health concern or condition. Participants integrated their provider's recommendation into their value, utility, and interest perspectives regarding PA.

"My doctor suggested that I lose some weight."

(FC-FG1-P6)

". . .eah, um, because uh we started off with a reference from a family Doctor. . ."

(C-FG2-P3)

". . .when I got my Parkinson's they recommended boxing or a punching bag, a speed bag, and um just the whole upper body movement, more upper movement, repetitive, sort of things."

(FC-FG1-P3)

*3.4. I Get Active with a Little Help from my Spouse and Others*

Being 'held to account' by others was a motivating factor, with participants describing how their partner or spouse sometimes provided a needed push when they did not feel up to engaging in PA.

"I think if you have a partner and if the two of you are really interested today I don't want to do something and my wife says okay let's do it. Tomorrow she doesn't want to do it, so we force each other."

(C-FG3-P3)

Similarly, being 'held to account' by knowing that other program participants or older fitness club members would be expecting them also provided the needed push, as summarized by one participant:

"I'm very good to giving into social pressures so if I know other people are coming and expecting me I'm more likely to go than if I'm just going on my own. I'm really good at making up excuses."

(C-FG3-P1)

*3.5. I See Changes and Improvements*

Participants described seeing positive changes in their health, fitness, and physical function through PA. Seeing improvements over time allowed participants to integrate the personal value and utility of PA, leading to continued and sustained engagement. This sub-theme was highly evident among the older adult fitness club members.

"Yes definitely, there is benefits to going. I know I was away from exercise for 3 years babysitting my grandson and I lost a lot of just my general strength and inabilities. I've been here for 5 months and I feel myself getting my strength back. I appreciate it a lot because I have heart issues and it's very easy to sit and you get very old and you just sit. This is good for me."

(FC-FG2-P5)

"I've been using a walker for 12 years I was young when I started and I'm just as active now with the walker in fact more active now than I was back then."

(FC-FG1-P2)

"Well, I think I'm a lot fitter than most of my friends to be quite honest." . . . Well, they've kept me fairly active. And able to do a lot more things than a lot of people my age."

(FC-IN1-P1)

"I always look at other people who are younger than I am and they can not even walk to the bus stop anymore. And I think that is an incentive for me to keep on moving."

(C-FG3-P5)

"Yes, I'm doing the best I can to improve and I have actually-the settings on resistance machines and that, they're going up. So I'm doing a little more than maintenance. And it takes a lot time of course to do-to get anywhere with this. But I think over the time I've made substantial progress. . .Well, the only real expectation I had was to get rid of the pain so I could sleep at night. Now, the fact that I'm sorta improving –I really wasn't expecting that to happen. You could say they more than met expectations."

(FC-IN2-P1)

Participants described several situations and factors that provided an incentive to begin or sustain PA or EX, highlighting participants' motivations along a continuum from extrinsic/external regulation to intrinsic/internal regulation.

Theme 2—*Facilitators and Barriers*

Many facilitators and barriers considered integral to participants engaging in PA (or not) were described. Facilitators included conditions, resources, processes, or support elements that enhanced older adults' ability to achieve their goals through PA and were related to the three SDT [30,31] psychological needs and underpinned participants' motivations. The sub-themes related to facilitators included: *I Can Do This*; and *Connections and a Sense of Belonging.* Conversely barriers (Table 2) were comprised of obstacles, challenges, limitations, and hindrances, with the sub-themes: *I Cannot Do This*; and *Pragmatics.* Environment- and interaction-related factors and other elements deemed either supportive (facilitators) or unsupportive (barriers, Table 2) of competence, autonomy, and relatedness were also discussed with the sub-theme: *Setting, Environment, and People Supports.*

**Table 2.** Barriers to Engaging and Participating in Physical Activity.

| Subtheme | Description | Representative Quotes |
|---|---|---|
| *I Cannot Do This* | This subtheme underscored the barriers identified by participants, specifically related to a lack of discipline as a fundamental hindrance to their engagement in PA. This challenge was particularly pronounced among participants who were not members of the older adult fitness club. The feelings associated with real and perceived lack of skills and abilities on sense of competence and, as a result, not finding PA pleasurable was unmistakeable. | "don't know where to start" (C-FG1-P1)<br><br>"Um, so, uh the problem was I think discipline . . ." (C-FG2-P3)<br><br>"I'm not disciplined enough." (C-FG1-P1)<br><br>"This is kind of a. . .it's my issue but I've never been engaged in sports. And I think part of it that I was never good at it. You know clumsy . . . You know you're. . .so I've never been involved in a positive way. Feeling that I can do those things. And I think in my head that's sort of been perpetuated so that I have joined gyms for years in the past. Like years and years ago and felt intimidated. You know sat on the equipment on the wrong way and had somebody come by and say oh my dear you're sitting on that backwards. You know like, you just kind of. . .reinforces that I'm not. . .I don't know if capable is the word but. . .so I never really enjoyed it." (C-FG3-P1) |

**Table 2.** *Cont.*

| Subtheme | Description | Representative Quotes |
|---|---|---|
| *Setting, Environment, and People Supports* | Participants who were not members of the older adult fitness club discussed a general lack of community-based PA programs and clubs for older adults, as well as limited aspects of programs and clubs for older adults that were in existence to address their needs and goals. The lack of optimal challenges and opportunities to engage in challenging activities to support the SDT competence need was evident. | "And um, I just find in XXX there's a lot of card playing people like that. . . .No but there's really nothing else that you know you can find." (C-FG2-P1) <br><br> "But that -sorry I was gonna say that that's there seems to be and maybe this is just me this notion that seniors just need the recreational piece of it . . . And-and I think through research and stuff." (C-FG2-P2) <br><br> "So, you're actually sitting and doing stretching and working with balls and stuff like that. . . and I'd like to do more than that." (C-FG1-P1) |
|  | Participants who were not members of the older adult fitness club described their negative experiences and lack of relatedness supports when seeking out PA or EX programs and facilities and centers which adversely affected their desire to join or participate. They were made to feel insignificant and not valued, and that they did not belong. | "I find the fitness club you go in as an old person and you still there looking after these guys over here. So that's the only thing I haven't really been in to look really, that's what I found really. They don't love old people." (C-FG3-P3) <br><br> "Yeah I have to agree with that. I looked into joining a club downtown. I got one of these free try us out passes. Nobody spoke to me, nobody looked at me. Lots of people looking at their own bulging muscles but. . .(laughter) . . .not at my flabby ones (laughter) . . . I would never return." (C-FG3-P1) |
| *Pragmatics* | Practical environment and setting supports were highlighted as important as they may be potential barriers to engaging in and sustaining PA: costs, e.g., need to be reasonable; convenience and location, e.g., not wanting to have to travel long distances, especially for those without a car or during inclement weather; need for small number of participants to allow for tailored attention, feedback, and recommendations for progression; and need for availability of safe and easy-to-use equipment. | "There wasn't parking in this lot, there wasn't parking across the street and I ended up having to go down to the school which is not a a huge problem but for someone who has ability issues it may be a problem . . ." (FC-FG1-P4) <br><br> "I'd like to see it open on Sunday for a few hours. I really regret that it goes from Friday to Monday." (FC-FG1-P5) <br><br> "And things like the um, swimming pool that's way up on XXX . . . It's far too expensive for me. . . You know? Um, so there are those limitations, it's either distance, bus, you know?" (C-FG1-P1) <br><br> "It depends on the fitness center or gym. It depends on the staff that they have. Now if they've got good staff you get lots of good training or whatever you need. Whether you're a senior or. . ." (C-FG3-P2) <br><br> "Yes and I don't want to travel really long distances to get to a place where I'm going to participate in something and I do. . .I understand all the reasons why it's beneficial to participate and be active. You know all. . .intellectually I've got it." (C-FG3-P4) |

PA = physical activity; SDT = Self-determination theory.

### 3.6. I Can Do This

The notion of "I can", highlighted the sense of capability and self-discipline required to meet competence and autonomy needs, ultimately fostering greater autonomous motivation. Feeling capable in general and more specifically believing they had the skills to engage in PA to achieve their desired outcomes and goals were very apparent among the older adult fitness club member participants. An essential aspect of competence with the SDT [30,31] is 'setting the optimal stage'—in other words, providing appropriate conditions and tasks that can allow the older adult to feel capable of engagement in the behavior and to demonstrate to themselves that they have the skills and abilities to engage in the behavior, as described by one of the participants.

"I was pleased with the program that was set up for me. Once it was set up I could administer it myself you know, 3 times a week."

(FC-FG1-P3)

In alignment with SDT [30–32], a sense of self-discipline and feeling capable and ready to engage in PA and enjoyment from engaging in the behavior are strong motivators and bring about integrated regulation.

### 3.7. Connections and Sense of Belonging

Relatedness-related reasons were evident through the participants' descriptions of feeling connected to others and a sense of belonging. Meeting and being with "like" others and socializing were important for participants. Interacting with others of a similar age group was preferred, allowing for the development of deeper connections. Participants noted engaging in PA or EX with others not only provided opportunities to socialize, but also enhanced their motivation and reduced feelings of isolation.

"Well I think the social aspect of it . . . Just getting out and meeting people and so on, you know and uh, I think that's a big part."

(C-FG2-P1)

"It provides motivation . . . like you say-walking-I mean I love to walk but it's better if you do it in a more social situation or where there's other people . . . Or even if you can talk with somebody."

(C-FG2-P2)

"I wanted to exercise, plus meeting people. I live by myself with a dog and a cat. Just meeting other people and talking to them. It helps I think."

(FC-FG2-P1)

"Yeah I think being around people you can joke with or talk a little bit, that's a big part of it."

(C-FG3-P4)

Participants noted the development of new friendships, the extension and expansion of their social connections and networks to other activities act as facilitators. . .

"I've made some good friends."

(FC-IN1-P1)

"I met my chess contacts through the Club."

(FC-FG1-P2)

"So. . . I think the incentive the ability to go and such and do things, meet people, have some continuity in your life is a good thing."

(C-FG1-P1)

### 3.8. Setting, Environment, and People Supports

Participants discussed various factors that assisted and supported them in undertaking PA and, conversely, that "put them off" (Table 2, Barriers) from doing so. Older adult fitness club members described a variety of club-specific features which were also key for the other participants.

Older adult fitness club members were more likely to describe autonomy, competence, and relatedness supports in general. Supportive behaviors were received from the health care professionals who suggested enrolling in the fitness club and from the fitness club staff. The older adult fitness club context provided competence support. The PA or EX program was developed in collaboration with the older adult based on assessment findings, individual and unique needs and abilities, and goals and preferred outcomes. This resulted in a program that appropriately challenged skill and experience levels and allowed for

building confidence. Providing opportunities to engage in their program independently based on the plan and the provision of non-judgmental informational feedback aimed at making improvements were additional competence support elements that were discussed by the fitness club members.

> "For my needs, it's been very adequate. I come down 5 days a week, we have a book in which they've listed what we should do so that we're not over doing it or, not doing enough and I found that worked very well, . . . they sat down and went through it so I knew what I was supposed to be doing, I wasn't guessing, because some of the them will write in the space what you're supposed to be doing. You don't understand what they're trying to tell you because they have a gym way of talking, so that you might not get it, but I found here it's very easy just to ask someone and they'll tell you, until you get on to your own program and then it sort of becomes, it's memory work."
>
> (FC-FG1-P6)

Autonomy was further supported through the staff being available to provide assistance and refraining from controlling or pressuring the fitness club members.

> "I think also when you ask for help, help is provided. They're not breathing down your neck all the time which is a good thing too. It's just finding that right balance. They've always been helpful."
>
> (FC-FG1-P7)

Autonomy support elements included: providing a rationale (e.g., to engage in EX) that was meaningful to the older adult and their circumstances and providing choices and options, e.g., the type of EX, activity, and equipment to use.

> "I was diagnosed with Parkinson's disease and they emphasized that exercise is vital to combat, to delay the onset of further Parkinson's problems."
>
> (FC-FG1-P3)

> "I think the choice of equipment is quite good. . . if you have to do strength or flexibility, it's just a matter of a hydraulic adjustment as opposed to taking off weights, that people might not have the ability to do and the mobility to move things around. . .it's just the matter of pushing a button. I think that is very very good."
>
> (FC-FG1-P9)

All participants described the importance of supportive environments, settings, and staff, and the critical need for staff to be aware of and knowledgeable about: the aging process; unique and specific needs of older adults; how the aging process and unique older adult needs transfer to ensuring safety; and how to best assist older adults achieve their goals and important outcomes through PA.

> "I think there's some issues about seniors trying to get fit that are quite different than people in their twenties, thirties and forties. I like to think that I've been collecting injuries for a very long time and I'm quite frankly nervous about starting a series of exercises or strengthening activities because I don't want to exasperate my pre-existing injuries. I know where my weaknesses are and I don't know how to get past them."
>
> (C-FG3-P1)

> "I think that's very valid. I'm feeling the same way. Like I want to know that someone is also, has expertise with this age group because of the limitations that we may experience with the aging process . . . I was more referring to the age specific changes that we might have."
>
> (C-FG3-P4)

"But they're not necessarily totally qualified so it's like. ... It's you, you don't have somebody who knows what they're doing."

(C-FG2-P2)

"You know I-I was fine before I got all this arthritis ... You know but then now I-I wanna go but I don't wanna go and be the only one that can't go like that ... And then the-the girl is so young, you know, and they try to understand but they don't understand."

(C-FG2-P4)

"if they can't tell me what I can do to help myself more I'm at a loss."

(C-FG1-P1)

The older adult fitness club members described the consistent positive regard of the staff towards them, and the warm and respectful interpersonal environment of the fitness club. They felt staff were genuinely interested in them, and that they belonged regardless of their physical limitations and skill and ability levels.

"And the staff here in-invariable say good morning to you. They're really good at it. It seems like a friendly place."

(FG2-P2)

"... the assistants who accompany those who have disabilities are also excellent and I've always been impressed with the quality of the personnel they have in the program."

(FG1-P2)

"Oh, I think that they're a very friendly bunch and certainly able to help if you help if you have a problem. Anyway, it's just a great atmosphere."

(IN1-P1)

The individualization of the initial PA prescription, opportunities for reassessment and progression, readily available individualized attention, and knowledgeable and helpful older adult fitness club staff were also discussed at length.

"Well they work with us individually too um, setting up the program we will follow we have books down in the gym that we keep track of what we do every day and what we accomplished um, it's a well-thought out program I think. There's something for everybody, not everybody does the same thing when they get down there, they more or less have their own programs."

(FG1-P1)

"They give you a program and they reassess you every few weeks, and that's an good, important thing to do to so you don't get stuck in the one set of exercises. Yup, yes they change [the program] as you get stronger or things in your life change." ... "Oh well, what I like especially, apart from the equipment. The equipment was good. But the people, the personnel, and they're great... they're tops. They help you they go around and they supervise. They're fantastic, that's it."

(FG2-P4)

"And I think they are very knowledgeable too. I've never had a question that they couldn't answer. I had both hips replaced and they were certainly helpful in my recovery from that."

(FG2-P3)

Participants who were not older adult fitness club members also discussed the need for and importance of the availability of assistance and supervision from qualified staff and programs specifically for older adults. Supervision during activities, receiving individual

attention, and having individual needs addressed were not available to them in community-based programs, fitness clubs, or centers.

> "...you are still going to need individualized assistance for you. Everybody else has got their own problems. So they have to look at everybody individually."
>
> (C-FG3-P2)

> "With the personnel to help with individual needs." ... "And help to get past [health limitations]. I know what the limitations are but i don't know how to get passed it."
>
> (C-FG3-P1)

Several facilitators and barriers were discussed by participants, including the need for supportive environments, settings, and staff, as well as more pragmatic factors.

## 4. Discussion

This study, which incorporated the SDT [30,31], explored the motivational factors that provide impetus and facilitate or hinder community-dwelling older adults' engagement in PA, within and outside an older adult fitness club context. The participants described a variety of intrinsic and extrinsic motivating factors along the SDT regulation continuum, and setting, environment, and people factors that did or did not support competence, autonomy, and relatedness needs.

We utilized SDT [30,31] to frame our research and deductive analysis, as we were specifically interested in motives and motivation. The theory and its related constructs were readily of value and relevance for categorizing, coding, and theming the raw data, apart from the *Pragmatics* sub-theme. SDT [30,31] is one of several theories, frameworks, and models that have been used to explain behavior and behavior change. Each has a particular underpinning, e.g., social cognition, humanism, dual process, and socioecological; as well as basic efficacy, strengths, and potential weaknesses when applied to PA [51]. Humans are complex and dynamic [51], as are the contexts and environments in which they live and function. One single theory may not explain all aspects of behavior, experiences, and perceptions, but theories are essential to enhancing the understanding of phenomena and providing guidance for interventions, programs, and policies.

A recent quantitative study incorporating an SDT approach of rural-dwelling individuals ages 65 to 96 years reported that enjoyment, appearance, fitness, health, and social engagement motives were significantly higher for those who exercised compared to those who did not, and individuals who engaged in structured EX scored higher for competence and relatedness, but lower for autonomy compared to those undertaking unstructured EX [52]. Appearance (extrinsic, a controlled form of regulation) was not a motive for the participants in our study. Our participants expressed enjoyment and satisfaction from PA, an intention to engage in PA, and the importance of PA for achieving health and well-being benefits, managing health conditions, and making social connections. Participants' need and desire to feel in control of and capable of managing their health condition, achieving health benefits and improving and preserving aspects of their physical function, such as mobility, strength, and flexibility, motivating them to engage in PA through the fitness club context or other settings, were highly evident. Disease management as a motive for EX has been reported previously in quantitative studies of community-dwelling older adults [53,54]. However in one of these studies, the older adult participants, aged between 75 and 81 years, cited mobility and poor health as barriers [53]. Pain, a fear of falling, and access have also been found to be central barriers to older adults' engagement with PA [55]. In contrast, we found that the older adult fitness center context provided effective competence and autonomy supports for the participants who were members despite health, mobility, and functional ability concerns and limitations. The setting and environment structures and processes were designed to be older-adult-friendly and the staff within the club had knowledge of aging and older adult needs, ensuring safety-related elements. The availability of easy-to-use and safe equipment, the individualized approach, and non-

judgmental attention and feedback from the staff helped participants become confident in their skills and abilities, allowing them to develop competence and feel ownership and capable of engaging in PA to achieve their desired outcomes and goals.

Ongoing community-based PA programs and facilities in non-health care and non-rehabilitation contexts, whether accessible through public or private funding, are cornerstone resources for addressing active and healthy aging policies and PA recommendations at the population level [2,4,6]. Participants who were not members of the older adult fitness club described negative experiences when searching for or trialing community-based PA programs or fitness clubs not specifically geared to older adults, and concerns about a lack of individualization, safety, and knowledge and skill levels of the staff. Not feeling sufficiently competent or skilled enough to engage in PA and a lack of effective competence, autonomy, and relatedness supports were limiting factors and barriers. Older adult fitness club members described the numerous club-specific setting, environment, and people factors that supported their needs and enhanced their motivation. This underscores the importance of more tailored programs, settings, and environments or resources to allow more individualized approaches to maximize older adult motivation and participation in PA and optimize skills and abilities to achieve outcomes and goals. While PA or EX toolkits, guidelines, program development resources for practitioners, and other resources do exist, 'older adults' are defined as those 50 years of age and older in many community-based programs. The skills, abilities, and unique needs of a 50-year-old or 60-year-old individual may differ from those in their seventies, eighties, and nineties. More targeted programs for older adults are often within the context of a health care setting or target a specific concern, e.g., risk of falls or health condition, e.g., cancer. More generic PA settings and programs catering to a wide range of age groups (including "older" age groups), although available, do not or are unable to undertake a more individualized approach; consider variations in abilities, skills, and preferences; or provide adequate supervision and supports. These limitations in meeting the specific and unique older adult needs related to autonomy, competence, and relatedness may result in less motivation or a lack of long-term and sustained engagement and participation.

Additionally, common barriers such as costs, convenience, location, and other more pragmatic factors were also described as important considerations, similar to what others have reported [24,56–58]. Although these pragmatics do not directly align with SDT [30,31] constructs, they can be considered as 'support' elements which have the potential to affect motivation and engagement. For example, if costs are limiting or the location is not convenient, the older adult may forgo a structured program or setting and they may or may not have the internal regulation to engage in no-cost, unstructured PA on their or with others, which in turn would affect the relatedness needs.

The participants discussed that their health care provider recommended PA, providing an initial impetus. Although older adults may factor in the personal value and utility of PA, these recommendations did serve as motivating factors. Although considered less autonomous within SDT [30,31], the suggestion from the health care practitioner was highlighted as playing a critical role in motivating participants. Cohen-Mansfield et al.'s 2004 quantitative study reported physicians advising EX, monitoring by a health professional, and the evaluation of the EX program by a professional were all rated as important or very important by at least 70% of the participants. A recent study reported that a vast majority (>80%) of health care professionals agreed they can play an increased role in promoting PA to older adults; however, only 30% indicated that they had received suitable training to initiate conversations about PA; less than a third had a clear plan on how to initiate discussions, and an assessment of PA was not routine [59]. Competence, autonomy, and relatedness support were found to be critical factors in our study, emphasizing the need for the knowledge and skill training of individuals who play key roles in PA and EX promotion for older adults.

Some participants also described being "held to account" by others as a source of motivation. While being "held to account" by others may be considered a more controlled form

of regulation, a 2017 systematic review found that social support for older adults: (1) was an important factor, particularly when the support is from family members; and (2) friends were important for engagement in leisure time PA [60]. Social support is important for the health and well-being of older adults [61–63], and behavior change theories emphasize the role of social factors including social support and social connectedness in initiating or maintaining behavior changes [64–66]. Furthermore, the strong desire to interact with others and feeling connected and a sense of belonging was highly evident among the study participants regardless of the participant group type, and it may be that being "held to account" by others, as described by our participants, may be more related to and reflective of having relatedness needs met.

*Strengths and Limitations*

Our qualitative study used a hybrid inductive–deductive approach to enhance the understanding of the phenomenon being explored. The inclusion of the SDT [30,31] to frame our research and deductive analysis allowed us to move beyond solely identifying motivational factors to garnering a deeper appreciation of the personal and contextual motivational factors that influenced older adult engagement in PA and in what ways, e.g., positively supported or hindered.

Including participants who were members of an older adult fitness club, older adults engaged and not engaged in community-based PA provided an opportunity for the rich exploration of similarities and differences, and in particular, the older adult club-specific support factors that contributed to the participants' competence, autonomy, and relatedness. These support factors have important implications for program design, implementation, delivery, and practice in community settings if the desired outcomes and goals of active living and active and healthy aging campaigns and policies are to be realized.

Our findings add to the 'older' adult PA motivation literature. Three quarters of the sample were participants between the ages of 70 and 89. We were only able to recruit three individuals aged 90 and older (two fitness club members and one not engaged in a PA or EX program). While the proportion of this age group in our sample (8.1%) was greater than the sampling city population proportion (1.0%), the perspectives and experiences of 'oldest' old individuals remain an ongoing gap in the literature that should be explored [67].

Despite undertaking multiple recruitment strategies, including non-digital and more traditional (flyers and posters in health and non-health centers accessed by older adults and various ethnic groups) to the use of social media, our sample was predominantly comprised of older adults who identified their ethnic origin as non-Indigenous North American or European. We were only able to recruit three participants who identified their ethnic origin as Caribbean and did not achieve the proportional diversity of the sampling city; thus, the findings may not be representative of generalizable perspectives and experiences.

While focus groups have long been used for social science and health care research, there are limitations to this data collection method such as the inclusion of self-selected participants decreasing the generalizability of the results, reliance on the facilitator to support and maximize the depth of discussions, and the potential for select participants to dominate discussions [68]. We followed best practices [69] including: group size, time limits, having trained individuals to facilitate the discussion, e.g., drawing out quieter participants and re-directing conversations, having trained observers present to take field notes, and audio-taping and verbatim transcribing of the focus groups and field notes.

Similarly, generalizability may be limited to our purposeful sample and additional contexts and elements, e.g., the geographic location, older adult fitness club, and other sample-specific characteristics such as the level of education and functional status. While we included participants who indicated interest but were not engaged in PA, we did not include older adults who were not interested in engaging in PA. Expressing interest in a behavior does not equate to an intention to engage in the behavior. A recent systematic review determined that while intention is necessary, it is an insufficient antecedent of PA for many—the intention–behavior gap was large and the likelihood of the successful

translation of a positive intention into behavior is low [70]. Frequently cited barriers among older adults who were not sufficiently active, beyond a lack of interest, are similar [71] to those we and others have found. Regardless, to what extent our findings may have differed had we included older adults who were not interested in PA is unknown.

Affordability and costs were described as barriers to participation in formal PA and EX programs and joining fitness centers and clubs. Our demographic data collection did not include a specific income-related question; thus, we are not able to report on the economic component of the participants' socioeconomic position. Rather we asked participants to indicate the amount they would be willing to pay on a monthly or per session basis to capture what was deemed acceptable for implementation consideration purposes. The evidence regarding associations between older adults' socioeconomic status and PA is insufficient nor definitively conclusive and more longitudinal research with unified measurement approaches is needed [72]. Lastly, while we found SDT [30,31] highly applicable and informative in describing and understanding our participants' experiences and perceptions with PA or EX, within and outside the context of an older adult fitness club, we acknowledge that SDT is one of many theories, models, and frameworks that could have been used as the basis for our work.

## 5. Conclusions

This qualitative descriptive study, using SDT [30,31] to frame the research, provides valuable insights into older adults' personal and contextual motivation factors, and key setting, environment, and staff support elements and factors related to engaging in and sustaining PA, within and outside the context of an older adult fitness club. The findings highlight the need for older adult-specific and tailored programs, settings, environments, and related components to optimize older adult motivation, their skills, and abilities through competence, autonomy, and relatedness supports to maximize and sustain PA.

**Author Contributions:** Conceptualization, V.D.B.-H.; Methodology, V.D.B.-H.; Validation, A.M., K.M., M.E.K., M.M. and V.D.B.-H.; Formal Analysis, A.M., K.M., M.E.K., M.M. and V.D.B.-H.; Investigation, V.D.B.-H.; Resources, V.D.B.-H.; Data Curation, V.D.B.-H.; Writing—Original Draft Preparation, A.M. and V.D.B.-H.; Writing—Review and Editing, A.M., K.M., M.E.K., M.M. and V.D.B.-H.; Visualization, A.M., K.M., M.E.K., M.M. and V.D.B.-H.; Supervision, V.D.B.-H.; Project Administration, V.D.B.-H.; Funding Acquisition, V.D.B.-H. All authors have read and agreed to the published version of the manuscript.

**Funding:** This research was funded in part from an Ontario Trillium Foundation grant sub-contract.

**Institutional Review Board Statement:** Ethics approval was received from the required Institutional Research Ethics Board (HiREB #5026).

**Informed Consent Statement:** Informed consent was obtained from all participants involved in the study.

**Data Availability Statement:** The dataset used in the current study are not available as we do not have ethics approval for this purpose.

**Conflicts of Interest:** Vanina Dal Bello-Haas was the recipient of the Ontario Trillium Foundation grant sub-contract. The Ontario Trillium Foundation grant was awarded to the organization that housed the older adult fitness club.

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
