# Peer review of "Engaging in and Sustaining Physical Activity and Exercise: A Descriptive Qualitative Study of Adults 65 Years and Older Using the Self-Determination Theory"

_2673-9259, doi:10.3390/jal4020011_

Round 1
Reviewer 1 Report
Comments and Suggestions for Authors
Line 82- not sure if this needs cited as it is your purpose?
line 89 guilt,
Line 121-123 either question marks after both questions or none at all
Line 185- a 12
Overall this is very well written and adds to the body of literature. The methodology is well explained and the introduction provides enough insight for the readers to follow along. The only suggestion I have would be to incorporate some of the conclusions into the abstract based on your findings instead of the broad topics covered in the discussion.
Reviewer 2 Report
Comments and Suggestions for Authors
This qualitative descriptive study examines the factors that influence participation in physical activity and exercise within and outside the context of a gym for older adults, using the Self-Determination Theory. The results reflect the existence of barriers and the lack of supportive environments, trained personnel and adequate programs.
The manuscript is clear and well-structured and the introduction is adequately elaborated. The materials and methods are clearly presented. The results are displayed logically.
Author Response
Dear Reviewer 2,
On behalf of the co-authors, I would like to thank you for your kind comments about our manuscript.
Vanina Dal Bello-Haas, PT, PhD
Reviewer 3 Report
Comments and Suggestions for Authors
This study has the potential to contribute meaningful empirical knowledge to an important field. However, I have some major concerns that need to be addressed:
Introduction:
There is no definition of PA and exercise in introduction. Considering exercise is a type of PA, there is no need to say ‘PA and EX’ each time in the paper. The authors could just ‘PA’, or if they are mostly concerned about exercise, they could just say ‘exercise’. This is a concern throughout the whole paper.
Lines 73-75: The authors say ‘few studies have explored individuals’ perspectives or have incorporated a theoretical framework as the basis for the research’ – Which population are they referring to? There is a wealth of qualitative evidence on individuals’ perspectives on PA/exercise across the general population, for older adults and for multiple health conditions (including older adult participants). There are also a lot of qualitative studies that use theoretical frameworks in PA/exercise research. The authors need to demonstrate much clearer knowledge and understanding of the existing qualitative literature for PA/exercise and provide a much more robust justification for their study.
Lines 77-79: What is the significance of the belonging to a fitness group/club? The authors mention that few studies have examined the potential influences of belonging to a fitness group – why might that be important and why is that important to explore? This is important as justification for the study.
The HC (health care) abbreviation is unnecessary throughout the paper.
The SDT paragraph is very long and doesn’t make sense where it is – it should be incorporated as a paragraph in the introduction and the introduction should finish with the aims of the study.
Methods:
The research questions in paragraph 1 would fit better in the introduction.
The authors state that they have used a qualitative descriptive methodology. However, they have used some explanatory methods by integrating theory into their results. I am not sure that this is the right description of their methodology, and the authors should consider reviewing this.
Line 161: The authors have said ‘at least 65 years of age and older’ – the ‘and older’ is not necessary here.
How did participants volunteer for the study?
Line 182: ‘participants provided written informed consent form’ – please revise this sentence, this is unclear.
Line 183: what demographic data and ‘other participant’ data were collected? Please explain and give examples.
There is no description about how the criterion/purposive sampling method was carried out to ensure a representative sample was achieved. The authors should provide this for clarity.
Section 2.3 paragraph 3: The authors have said that all participants returned consent forms prior to the phone call/interview/focus group, and then say there were two observers that helped the participants with the consent forms and questionnaires – this seems contradictory – if they completed the forms before the call started, why would they then get help with these during the call? Please clarify and correct in the manuscript.
How was the interview guide created? E.g., was it from literature, using a framework or subject experience? And was the interview guide piloted or changed as the data collection progressed?
Reflexivity statement – were the participants aware of the researchers position prior to data collection? E.g. did they know the background of the researchers? Please clarify and outline in the paper.
Data analysis: The authors have described the inductive process, but it is not clear how the inductive and deductive processes link together – please amend to make this clearer how the SDT came into the analysis?
Data analysis: did the authors use an existing/published thematic approach for their inductive analysis: if so, which one? If not, why not?
Results:
Paragraphs 1, 2 and 3: this would be much more readable in table format. I recommend the authors shorten the sentence to refer to a table and then put these results in the table.
Why are some results in Table 1 but the others are written in the text – please make the results consistent to help with readability.
The results generally are very long – it would improve the readability of the paper to make paragraphs more succinct and reduce the words.
Discussion:
The first sentence is incorrect – there are many qualitative studies that use theory – please see comments on introduction.
There is little discussion on the SDT. This is a potentially valuable part of the paper as the authors could discuss how the findings support the SDT as a theory explaining PA for older adults and where there were also non-SDT factors (if any) and where the SDT may or may not be the best theory to explain PA behaviour. There are many theories that have been used to try and explain PA behaviour and the discussion would be richer if it included a section discussing this. See Rhodes RE, McEwan D, Rebar AL. Theories of physical activity behaviour change: A history and synthesis of approaches. Psychology of sport and exercise. 2019 May 1;42:100-9.
A potentially significant limitation of this study is that the majority of participants were engaged in PA and the participants that weren’t engaged in PA were interested in engaging. This suggests a predominantly high level of motivation and engagement and even those not engaging had positive intentions to do PA. This is likely to affect the results as key barriers are likely to be missed that will be present for those not engaging or not intending at all. The authors should discuss the implications of this on their results and conclusions.
The authors should discuss the limitations of using predominantly focus groups to collect their data. Whilst focus groups are a useful method for several reasons, they can limit the depth of the collected data.
Many people do PA that is not in community/club environments, such as at home, going for walks, hiking or playing a sport. This paper is largely geared towards community-based PA in a fitness centre/club. The authors should discuss the implications of their results on the population of people that are doing PA in a variety of different environments to discuss generalisability.
Comments on the Quality of English LanguageThis paper is generally well written. However, it is a little long in places, particularly the results, and could be written much more succinctly.
